# Is There Somebody Looking out for Me? A Qualitative Analysis of Bullying Experiences of Individuals Diagnosed with Bipolar Disorder

**DOI:** 10.3390/ijerph21020137

**Published:** 2024-01-26

**Authors:** Gülüm Özer, İdil Işık, Jordi Escartín

**Affiliations:** 1Institute of Psychiatry, Psychology & Neuroscience, King’s College London, London SE5 8AF, UK; jordiescartin@ub.edu; 2Department of Psychology, Faculty of Economics, Administrative and Social Sciences, Bahcesehir University, Istanbul 34353, Türkiye; idil.isik@bau.edu.tr; 3Department of Social Psychology and Quantitative Psychology, University of Barcelona, 08035 Barcelona, Spain

**Keywords:** bipolar disorder, workplace bullying, qualitative study, deductive content analysis, organisational psychology

## Abstract

According to the World Health Organisation, there are globally 40 million individuals with bipolar disorder (BD), and they experience stigma and discrimination, as many people with mental illness do. Work bullying (WB) is a common organisational problem, deteriorating the well-being and performance of employees and organisations. Although WB experiences have been researched for over three decades, we do not know much about the experiences of this group and what they need to extend their work-life. The current research aims to understand the workplace bullying experiences of individuals with BD and factors that may foster or hinder their participation in the labour force. The research methodology was based on in-depth interviews with 19 employees diagnosed with BD. Bullying experiences were mapped onto the Negative Acts Questionnaire. The data were analysed using the deductive qualitative content analysis on MAXQDA. Results showed that individuals with BD were exposed to bullying in work settings. Perceived reasons for the negative behaviours were mainly the undesirable individual characteristics of the bully, prejudices toward bipolar disorder, and already-existing toxic behaviours within the organisation. To reduce bullying, participants suggested that, among others, top management include equal and fair treatment of disadvantaged employees as performance criteria for supervisors and that organisations provide training against bullying, investigate complaints, apply sanctions, and establish an inclusive environment. If organisations set a stable and enduring vision, enhance a common identity for all employees, work on team building, and emphasise team efforts and goals, the organisational environment would be more inclusive, and individuals with BD would have longer work lives.

## 1. Introduction

Amidst the complexity of bipolar disorder, there is a struggle often overshadowed by the ups and downs. This struggle is further compounded by the effects of potentially toxic interpersonal relationships in work settings, namely, through the global burden of bullying. A question echoes through the corridors of these personal battles: Is there anyone looking out for me? In this research, we conduct an in-depth examination of the bullying experiences of individuals diagnosed with bipolar disorder. Beyond the clinical lens, this qualitative analysis delves into the bullying experiences of individuals with this psychological disorder, shedding light on the often-overlooked emotional domain. The current research aims to illuminate untold stories at the intersection of psychological health and interpersonal difficulties.

Previous research on employees with psychological illnesses shows they experience bullying, a high-intensity negative behaviour [1]. Bullying is operationally defined as hostile and unethical systematic communication directed towards an individual over six months with at least one negative act per week [2]. Regarding duration, researchers make definitions such as “long-lasting” or “at least six months”, leaving out one-off conflicts [3]. When analysed by method and geography, bullying prevalence in European countries is 4.6% to 29% according to the self-labelling process and between 4.1 and 44.8% according to the operative criterion of “one act at least once a week for at least six months” [4].

Scholars have theorised that bullying stems from (i) work conditions such as role conflicts, work overload, and job ambiguity created by poor job design and an unfavourable social environment (The Work Environment Hypothesis [2]); (ii) the individual characteristics of the target or perpetrator such as emotional and behavioural reactions and lack of coping strategies [3]; or (iii) combination of both factors. These factors combine, forming destructive organisational settings with escalated conflicts and ineffective coping [5]. Previous research on work environment factors showed that role conflict, workload, role ambiguity, job insecurity, and cognitive demands predict being a target of workplace bullying [6]. Additionally, absent leaders who do not manage critical situations are instrumental in conflict escalation and increasing bullying incidences [7,8]. Researchers working on individual differences showed that bullying may stem from targets’ traits, such as anger, neuroticism [9], anxiety [10], and low conscientiousness [11]. Bullying cases may start as work conflicts and progress subtly, where the individuals may not realise they are being targeted or face direct negative behaviours. Targets may be exposed to work-related bullying, such as demanding unmanageable workloads, excessive monitoring, and being assigned meaningless tasks, or person-related bullying, such as being insulted and teased and being exposed to gossip, rumours, outrageous jokes, and criticism [12]. These negative behaviours inflict stress and helplessness, cause psychological distress, suicidal ideation, depression, anxiety, sleep difficulties, headaches, bodily pain, and cardiovascular diseases. As a result, targets may incur sickness absence or come to work feeling sick (presenteeism) and request disability pensions [13]. Therefore, one of the main risks for employees at workplaces that may cause psychological or physical harm is workplace bullying [14]. The adverse effects of bullying on human health and well-being are further exacerbated by a vicious cycle, leading to progressively poorer mental health [15].

According to the World Health Organisation, one in six working-age people has a mental condition. This situation highlighted the gaps in national capacity to support mental health in the workplace, where only 35% of countries reported having government programs to promote and prevent mental health at work. Employers have a substantial financial reason to encourage their employees’ mental health, yet investment in mental health is low, and the stigma associated with mental illness persists [16]. Globally, there are 970 million people with mental disorders, of which 40 million have bipolar disorder (BD) as of 2019 [17]. Previous studies have indicated BD prevalence rates as 1.72% in the UK (*n* = 7546; [18]) and 3.7% in the USA (*n* = 85,357; [19]), with other studies reporting global prevalence rates of 2% [20] and 5% [21].

### 1.1. Navigating Bipolar Disorder in the Workplace: Challenges, Accommodations, and Societal Attitudes

The Diagnostic and Statistical Manual of Mental Disorders [22] distinctly categorises BD into two types. Bipolar 1 disorder is marked by severe manic episodes, characterised by highly elevated mood and energy, impulsive behaviour, decreased need for sleep, and racing thoughts. These episodes, lasting at least a week, can significantly disrupt daily life or require hospitalisation. Major depressive episodes may also occur but are not essential for a Bipolar 1 diagnosis. Bipolar 2 disorder, on the other hand, involves a pattern of depressive episodes and less severe hypomanic episodes, lacking the full-blown manic episodes of Bipolar 1. The depressive episodes in Bipolar 2 are often more prominent, and the milder hypomanic episodes, which last at least four days, do not significantly disrupt daily functioning, setting Bipolar 2 apart from Bipolar 1 [20,21]. Various effective therapeutic methods are available for treating BD, including medication, dietary supplements, stress and conflict management, counselling and psychotherapy, and social functioning enhancement [20]. The abnormal mood states of BD are accompanied by mental and body functioning changes, such as problems with motivation and energy levels and decreased focus and memory [20]. Work impairments after the onset of illness may be due to (i) older age and lower education levels; (ii) decline in cognitive functions; (iii) comorbidity of BD with other conditions (e.g., substance abuse, personality disorders, anxiety); (iv) adverse course of the illness (e.g., the high number of hospitalisations, high number/severity of symptoms); and (v) depressive symptoms. Work and social function impairments may result in high unemployment for this group, such as 20–58% in the USA, 61% in Canada, 28–61% in Sweden, and 62% in Denmark [23]. Despite the impairments caused by the illness, rapid developments in diagnosis and treatment methods have enabled many individuals with BD to return to work. Consequently, BD has been included in the disability acts worldwide to protect these employees [24,25,26]. A recent study showed that after the initial onset of the illness, 34% of individuals with BD returned to work after five years [27]. However, they incurred lost workdays [28] and showed adverse performance, behaviour, or mental attitude regarding feelings and beliefs [29]. It is also worth noting that individuals with BD form a heterogeneous group, and a non-negligible percentage maintain good social functioning [30]. Up to 60% of individuals with BD are employed over the longer term despite demotions [31]. They stay employed if they have good verbal memory [32], have higher education levels [33], have flexible jobs and workplace support [34], and their attacks do not lead to hospitalisation.

Effective implementation of disability-related labour laws has been crucial as employment for this group enhances their recovery [35], improves daily rhythm and well-being [36], fosters financial benefits, pride, and self-esteem [37], and lowers suicidal risk [38]. The employment of this population is also vital for national economies. Costs including unemployment payments, medical and hospitalisation expenses, productivity losses of employees with BD, and the health costs of the caretakers were calculated to be USD 195 billion annually in the USA in 2020 [39]. Regrettably, individuals with mental illnesses in the workplace often encounter negative behaviours. They face stigmatisation [40], avoidance, and discrimination [41], resulting in less disclosure of their conditions, reduced recognition of their rights, and consequently, shorter work lives [42]. A literature review revealed that mental illness is sometimes perceived by others as a lack of character or deficiency, leading to diminished social support, decreased occupational success, impaired functioning, exacerbated symptoms, and lower quality of life [43]. Individuals with BD are too often regarded as untreatable, unpredictable, dangerous, unreliable, and incompetent by society [20]. A recent study showed that when subject to workplace mistreatment (such as incivility, social rejection, and ostracism), they report suicidal thoughts [44].

### 1.2. Aim of the Current Research: Underexplored Risk of Bullying against Employees with Bipolar Disorder

The research on the in-depth effects of bullying on employees with BD is limited, as mainstream bullying research has not focused on this group. However, bullying poses a significant risk to employees with BD, jeopardising this population’s employment, as is the case for other disadvantaged groups. This research sheds light on a relatively underexplored topic with great potential for further investigation. In particular, qualitative studies that unravel the bullying experiences of this group are essential to understanding how this hostile conduct may affect them [42]. A previous review indicated that the BD prevalence rate in Türkiye is 1% [45]. In contrast, the prevalence rate of workplace bullying in Türkiye seems to be much higher than European rates, with wide fluctuations due to the different scales used and limited sample sizes, 63.4% among junior physicians (*n* = 394; [46]), 90.1% among bankers (*n* = 213; [47]), and 36.9% among healthcare workers (*n* = 103; [48]). Literature reviews indicate that in instances of bullying, supervisors are the perpetrators about 50% of the time, and colleagues 42.5% of the time over research conducted with the general employee base [49]. Particularly, in high-power distance countries like Türkiye, bullying is often seen as a form of power abuse by superiors, which reinforces the findings [50,51].

To our knowledge, bullying research on employees with BD has also been limited. We do not have an in-depth understanding of this demographic’s bullying experiences, the challenges they confront, or the assistance they receive at workplaces to maintain their employment and protect their well-being. Therefore, through a qualitative exploratory approach, this study aimed to examine the workplace bullying experiences of individuals with BD. The objectives were to explore the types of negative acts encountered, perspectives on the causes of bullying, experiences during and after the bullying process, and suggestions for preventing bullying. Additionally, the study questioned what reasonable workplace accommodations could ensure equal conditions to sustain employment. This study aims to identify measures for preventing bullying against individuals with BD and to guide human resource practitioners (HRP), organisations, and policymakers in creating work environments that support their well-being.

## 2. Methods

This study used a deductive qualitative research methodology with semi-structured in-depth interviews to explore bullying experiences and to gain insights into establishing bullying-free work environments and keeping individuals with BD in the labour force. The study features a purely qualitative design, devoid of experimental elements. Human Research Ethics Committee Approval was obtained from Istanbul Bilgi University dated 25 July 2019.

### 2.1. Participants

The sample (*n* = 19) was diverse in terms of gender (female: 42%), age (average age: 35.6 years; SD: 8.9), and tenure in work-life (average tenure: 12.2 years; SD: 7.3) (Table 1). The sample size is acceptable following Henning and Kaiser’s (2022) systematic review [52]. In their study, they reviewed 23 articles on data saturation in qualitative research, and results showed that data saturation was reached within the range of 9 to 17 interviews. In the current study, participants worked for public and private sectors: 37% for government agencies, 26% for manufacturing, and 16% for retail. Participants’ average age when diagnosed was 27.4 years (SD: 10.67), and they had worked 3.2 years on average in their current organisations. Nine participants held an undergraduate degree, five had graduate degrees, three had a pre-undergraduate degree, and two held a junior high school diploma. Participants’ professions, such as strategic planning manager, clergyman, social worker, construction worker, and food engineer, were highly diversified. Twelve participants were currently employed, two were retired, two were unemployed but not actively seeking employment, and three were unemployed but actively seeking employment. Twelve individuals (63%) disclosed their diagnosis at work. Thirteen participants reported experiencing bullying, whereas the remaining respondents characterised their encounters as negative behaviours that did not entirely align with established bullying criteria. Four participants mentioned that they have an anxiety disorder (Participants 3, 7, 14, 16). Anxiety disorders can exacerbate emotional sensitivity and cognitive perception issues in individuals with bipolar disorder. People with bipolar disorder, already prone to emotional fluctuations, may find these acts intense because of the co-occurring anxiety. Anxiety can also affect cognitive functions, leading to heightened or altered perceptions, especially during manic or depressive episodes [20,21]. Therefore, even though the coding of interviews within this group was completed, the coded segments were omitted from the final manuscript in terms of representative quotations.

### 2.2. Participant Recruitment Process

The study was announced on Türkiye’s Bipolar Disorder Facebook group. Forty-nine individuals contacted the first author expressing their interest in participating. Before conducting the interviews, all volunteers received a call during which they were briefed about the research. We communicated the inclusion criteria as follows: (a) being in remission, with no depressive or manic attacks in the past three months; (b) having work experience, including part-time jobs (minimum 8 h per week); and (c) signing the consent form. Thirty participants did not meet the inclusion criteria: twenty-one were not in good health; six had not worked yet, and three did not sign the consent form. Consequently, 19 participants joined the study.

### 2.3. Interview Protocol and Process

In October 2019, a welcoming environment was created for semi-structured, in-depth interviews in Turkish with 19 participants. Face-to-face (*n* = 3) and telephone (*n* = 16) interviews were conducted depending on the participant’s choice following the supporting literature on the effectiveness of telephone interviews in clinical research [53]. The discussions were based on open-ended questions designed to reveal the bullying experiences and the impact of these experiences on them. The interviews were intended to be informal conversations [54] so that participants would feel comfortable discussing sensitive topics with the researcher. Research [55] highlights the critical role of the interviewer in encouraging the participants to disclose their subjective experiences through open communication. Rather than a structured set of questions, we prepared a list of areas that the researcher wanted to cover as a guide.

Initially, we collected demographic information by having participants introduce themselves. We inquired about their current employment status and their diagnoses. We mentioned that workplaces could have a wide range of negative behaviours, starting from simple incivility to full-blown physical assault. Bullying was defined as repeated exposure to unpleasant or degrading treatment that individuals find challenging to defend themselves against [56]. We requested that participants describe the negative acts they had encountered at work, how these acts impacted them, and how they had coped. Subsequently, we delved into detailed questions about the impact of bipolar disorder (BD) and bullying: “How does being diagnosed with bipolar disorder affect your daily life and work life?”; “Have you directly experienced bullying in your professional life? If so, how, when, and by whom? Can you describe the process and its effects on you and your work life?” Participants were also prompted to reflect on their experiences by exploring the reasons for bullying and suggesting potential interventions to address it. Lastly, we sought participants’ recommendations for organisations and policymakers to better support individuals with BD in the workforce.

### 2.4. Procedure and Data Analysis

With the participants’ consent, we audio-recorded the interviews, which lasted between 16 and 74 min, averaging 39 min. The total interview duration was 741 min. We conducted a follow-up with the participants one week after the interviews to assess any shifts in their moods. They responded that no changes had occurred, and they affirmed being in good health.

After transcribing the audio recordings of the interviews, we performed the deductive qualitative content analysis [57] on the transcripts using the MAXQDA data analysis software (version 2018.2). Research [58] emphasises the importance of incorporating theories in qualitative research, particularly when using robust and widely recognised approaches, allowing researchers to “discover their voices” [59].

Accordingly, in the current research, Einarsen’s (2000) [3] framework of negative acts guided the deductive coding process. Participants’ experiences were mapped against the Negative Acts Questionnaire (NAQ-R [60]) to determine what types of negative acts participants were exposed to. Systematic reviews established that this scale is a reliable and thus a frequently used instrument in determining bullying in different cultures [61,62,63]. For negative acts to be considered bullying, unwanted behaviours must occur systematically and persistently, where the targets cannot stop the behaviour [64].

Alongside the list of negative acts, we created a predetermined coding scheme to identify and map the antecedents of bullying experiences based on available literature [65]: negative behaviours in business life, reactions to negative behaviours and their consequences, the perceived reasons for negative behaviours, suggestions to eliminate negative behaviours, and conditions necessary for the existence of individuals diagnosed with BD in business life. However, the addition of the new themes was allowed. Once results were obtained, the segments were translated into English to be included in the manuscript.

In presenting the study’s results, we took care to protect the participants’ identities, replacing names with participant numbers and referring to their workplaces by sector rather than by specific institution.

## 3. Results

### 3.1. Various Forms of Negative Behaviours

The interviews depicted sustained negative behaviours towards individuals with bipolar disorder, which can be categorised as bullying, as shown in Table 2.

Participants reported experiencing negative behaviours that persisted for months, and in some cases, even years. According to the scale used, the most frequently encountered negative behaviours included being insulted, excluded, and assigned unmanageable workloads. They noted that the primary sources of bullying were their supervisors and peers. Concurrently, Human Resources Practitioners (HRPs) played a significant role in terminations and demotions, often in collaboration with the top management of the companies. The results indicate that participants were subject to bullying due to, and in spite of, their condition.

### 3.2. Underlying Causes of Workplace Bullying

As illustrated by the sample quotes in Table 3, participants identified the toxic work environment as one of the key factors contributing to bullying. For instance, Participant 11 stated, “Bullying is a learned behaviour. This is how they learn from others. They continue to practice it, either unconsciously or knowingly, like a tradition”. Another participant’s experience with the disability report process revealed another aspect of a toxic social environment: “I was trying to get a disability report. Later, I found out that this was a topic of teasing in the company” (Participant 18). As Participant 10 pointed out, “You inevitably get disturbed by people’s conversations among themselves, their choice of words, their perspectives on events, their comments”. Such negative social interactions created an unhealthy context.

Furthermore, leadership was identified as a crucial factor. Participant 11 highlighted workload issues: “He was constantly putting pressure on us to finish the task at hand. We were constantly stressed. He was verbally abusive”. Participant 4 observed, “If leadership is inadequate, anything could happen”. Similarly, Participant 5 commented on the leaders’ impact as role models: “Fish stinks from the head. If you observe ridicule (as the supervisor) and you do not intervene, then this person (referring to herself) is exposed to all kinds of mobbing”. In such environments, dismissals are not uncommon, as Participant 15 experienced: “I went to the company doctor and disclosed my diagnosis after joining the company within 1.5 months. The doctor talked to the management. After that, they dismissed me immediately”.

The second main theme related to individual differences, emphasising the characteristics of the bully. Bullies were often perceived as frustrated and dissatisfied with their own lives, leading them to target others as a means to vent their emotions. Participant 19’s statement underscored how a lack of clear personal goals and job dissatisfaction contribute to a bully’s authoritarian behaviour:

“The bully doesn’t have his own goals. He chose a profession he doesn’t like. He works at a job he dislikes or has no respect for his work. He finds pleasure in torturing people and being authoritarian. Or he’s just bored and has nothing else to do.”

Ego issues, ruthlessness, and viewing bullying as a power game were various psychological facets of the bully’s mindset, as highlighted by Participants 6, 18, and 2. Insecurity among managers may led them to target employees they perceive as threats, reflecting the power dynamics within the organisational structure. Participant 15 elaborated on this:

“It depends on the character of the person. If there is no problem with his character, he already understands how the disease impacts us. However, if a person’s character is flawed, he will engage in bullying. He can mock others’ flaws, regardless of whether they have bipolar disorder, are healthy, or physically disabled.”

These suggested that bullying may be ingrained in an individual’s character or may represent learned behaviour, hinting at a broader organisational culture (Participant 6, 11).

The analysis also revealed the causal themes related to the victims’ perspectives. Individuals with bipolar disorder express challenges in conforming to societal expectations, emphasising the authenticity of their emotions (Participant 9). Being perceived as different, especially when exhibiting a colourful personality, may make individuals targets for bullying (Participant 10). Similarly, Participant 5 mentioned that “You are different. You are not like them. During depressive periods, your work slows down, and you are always sullen. You will be humiliated because you are different.” The link between depressive periods slowed work pace, and humiliation suggests that perceived differences contribute to victimisation (Participant 5).

In summary, the analysis underscored the intricate interplay among individual characteristics, organisational dynamics, and the unique challenges faced by individuals with bipolar disorder in the context of workplace bullying. The findings offered valuable insights for understanding and addressing these issues, both at the individual and organisational levels. Participants reported that bullying occurred under abusive supervisors and absent leaders, where negative behaviours were not condemned but replicated. They generally portrayed perpetrators as cruel, self-centred, narcissistic, having low self-esteem, and lacking empathy. Participants believed that perpetrators either disliked or disrespected their jobs, acted out deliberately as part of their strategic career plans, or engaged in such behaviour due to social learning in the workplace. Participants believed that individuals with BD would be more bullied due to their different behaviours or moods, especially during their depressive episodes.

### 3.3. Impact, Strategies, and Consequences of Bullying

The experiences shared by employees with BD who faced bullying underscored the pervasive stigma and social isolation at the workplace. Participant 1 highlighted the prevailing scepticism and discrimination, noting the belief that individuals with bipolar disorder cannot achieve success in various aspects of life, leading to social withdrawal and harmful stereotypes by saying “They do not think that bipolar people can be successful in their social life, business, or love life. And they inevitably withdraw. We still come across such things as ‘he is a bit crazy, he is taking drugs, be careful’.”

Participant 15 expressed exhaustion from the strain on personal relationships caused by societal misconceptions, emphasising the impact of societal attitudes on individuals managing bipolar disorder:

“The deterioration of human relations is tiring for me. In any episode, relationships are instantly broken. Episodes come and go if you manage them correctly. I have a disease. Treatment methods are clear. What makes me tired is that the social environment is affected too much. People leave me.”

Participant 5 provided a poignant account of social exclusion during depressive phases, where colleagues criticise their personality rather than their professional abilities: “With work colleagues, we go to entertainment places. If I am in a depressive phase, I sit sullenly. They exclude me. People always criticised me for my personality, never about my job.”

These narratives collectively illuminated the profound influence of stigma on interpersonal relationships in the workplace, revealing a need for increased awareness and education to foster an inclusive and supportive work environment for individuals with bipolar disorder.

This context of being a victim of bullying has emotional impacts like sadness and shame, to list a few from the Figure 1. Individuals with BD mentioned that they blamed themselves while exposed to bullying. Their physical and psychological health deteriorated (e.g., migraines, fibromyalgia pains). Most importantly, bullying triggered manic or depressive episodes related to BD for some participants, as the left circle in Figure 1 shows.

We observed that participants employed different strategies or changed their strategies while being bullied. Participants 10, 15, 6, and 12 offer various coping strategies, including communication with managers, setting boundaries, addressing the issue directly with bullies, and choosing not to respond to bullying behaviour, as the following segments depict:

“If the situation bothers me a lot, I go talk to my manager, ask their opinion. If still, things do not get back to normal, then I file a complaint, I do not hesitate.” (Participant 10)

“You need to draw your boundaries as much as possible.” (Participant 15)

“Tell the bully that it is wrong to act like this, that they should put themselves in the other person’s shoes, and empathise. It is best to have open communication and solve it.” (Participant 6)

“Do not respond to bullies.” (Participant 12)

These statements clearly indicate that participants reacted by not taking any action or by staying silent and ignoring the bully. Some confronted them and filed a complaint if the bullying did not stop.

However, some participants resigned due to bullying. Three participants continued to work at their organisations while undergoing bullying. Their complaints to human resources did not stop the bullying they experienced (Figure 1).

### 3.4. Empowering Strategies to Combat Workplace Bullying towards Individuals with BD

#### 3.4.1. Organisational Solutions

The suggested organisational solutions encompass the importance of effective leadership, establishing supportive policies and procedures, comprehensive training for supervisors, and the enhanced understanding of human resources management professionals in creating a workplace environment that reduces bullying towards individuals with bipolar disorder.

(a)Public Training and Hotlines: Participant 11 recommended anonymous reporting through government hotlines, emphasising the need for confidentiality to encourage reporting of bullying incidents: “I think you should be able to report anonymously to the government’s hotline, but they ask your name. Unions can do it.” (Participant 11)(b)Organisational Audits: Participant 18 suggested governmental intervention through organisational audits to ensure the well-being of employees with bipolar disorder, highlighting the importance of state protection:

“If the state does not protect its citizens diagnosed with bipolar disorder, I do not think another person will defend their rights or treat them well. The state should visit workplaces to check on bipolar employees following up on their work conditions.” (Participant 18).

(c)Mental Health Checks and Awareness: Participant 9 proposed incorporating mental health and disability tests into public health screening from early childhood, stressing the importance of early detection and support for mental health issues.

“Doctors come to schools for vision screening when we are kids. However, psychiatric and psychological tests are not done. Until military service, I did not know about my condition. Public health screening should include psychiatric disability tests from early childhood.” (Participant 9).

Bullying was often experienced due to disclosing the diagnosis. Some participants believed it was essential to hide their BD diagnosis as they were exposed to ridicule by their colleagues and supervisors after they shared their diagnosis. However, most participants suggested that individuals diagnosed with BD be open about their diagnosis and create awareness in others to reduce workplace bullying and obtain much-needed support. Participants emphasised the importance of psychoeducation on mental illnesses in general and on how to prevent bullying. They urged the state to support employees with BD, allowing for anonymous complaint filing on bullying hotlines and revising the law to protect the victims further. The need to raise awareness of acts of bullying was underlined. Participants also advised other victims to use different coping strategies, such as initially ignoring the bully, then voicing their concerns to the bully, and then filing a complaint with management if necessary.

Participant 7 underscored the importance of training others, i.e., psychoeducation, about bipolar disorder by likening it to a physiological condition, emphasising the use of medication for emotional balance: “This is a condition. Tell them. Just like a diabetic suffering from insulin deficiency, I have a lithium deficiency. I am using that drug; my balance is coming back. That is, it!” (Participant 7).

As Participant 10 notified, “I did tell (the bipolar diagnosis), but I am not sure if they knew what the disease meant. I tried to explain” training on mental illnesses emphasises the need for awareness and understanding, both for colleagues and employers, fostering an inclusive and supportive workplace as Participant 7 described as well: “I talk about my illness in advance. They welcome me with understanding. I have never hidden it. Everyone should know about this illness.” Participant 11′s caption, “Most importantly, the side effects of drugs should be known”, is also an essential highlight.

(d)Leadership: Participant 4 emphasised the importance of effective leadership, suggesting that organisations seek leaders who can create a positive and harmonious environment for all stakeholders: “Find good leaders like orchestra chiefs, that could make all stakeholders happy.” (Participant 4).

Participants’ suggestions for organisations centred on sound leadership, setting up policies and procedures against bullying and training employees, especially leaders, on mental illnesses. Participant 19 underscored the need for comprehensive training for supervisors, focusing on mental illnesses and strategies to address workplace bullying effectively: “Supervisors should be trained in-depth on mental illnesses and bullying at work.” (Participant 19).

Moreover, as Participant 5 highlighted, the importance of being well-informed about bipolar disorder is critical for human resources professionals as well, emphasising the need for awareness in hiring and retaining employees with bipolar disorder: “Human resources directors need to understand what bipolar is. They need to know what to expect when hiring an employee with bipolar disorder.” (Participant 5).

(e)Policies and Procedures: Participant 1 recommended organisational policies that foster communication, performance evaluations, and strict consequences, including the possibility of punishment or dismissal, for workplace bullies: “Organisations should establish an open communication environment, frequent performance reviews. Bullies should be punished or even dismissed.” (Participant 1).

#### 3.4.2. Self-Employment or Public Sector Employment

Participant 5 recommended self-employment or seeking employment in the public sector, where legal rights provide protection against discrimination based on bipolar disorder:

“If you say ‘I am bipolar’ before you get hired, you will not be hired. If you say, ‘I am bipolar,’ after being hired, they will start looking for ways to fire you as soon as possible. I advise people with bipolar to choose jobs they can do themselves and be self-employed. Or get a job in the public sector. Whether you are bipolar or schizophrenic, they cannot fire you after the state hires you. Because you have serious legal rights.” (Participant 5).

In summary, these victim-centric solutions proposed coping strategies, employment choices, and education as key tools to reduce bullying towards individuals with bipolar disorder. The recommendations focused on empowering victims to navigate and address workplace challenges effectively.

#### 3.4.3. Suggestions on Accommodations for Long-Term Employment

There appears to be a downward drift in occupational status over time for individuals diagnosed with BD. We inquired about workplace accommodations needed to keep individuals with BD employed longer. All the participants contributed to accommodations for long-term employment. The most frequently mentioned suggestion was to create a supportive environment for this population where the diagnosis can be shared without shame and fear, and managers and colleagues would know about the illness and understand the individual’s difficulties. Participants emphasised the importance of having flexible work schedules and having time for health visits. Other organisational accommodations mentioned that would help individuals BD to stay employed longer were job redesigns allowing for frequent breaks, organisational training on mental illnesses to reduce possible stigma, and recognising the supervisors for good performance in managing this population. One participant mentioned that the government could establish sales channels for individuals who would choose to be self-employed.

(a)Creating a Supportive Environment: Open communication about bipolar disorder, understanding, and recognition of the unique challenges faced by employees with this condition is critical as Participant 8 mentioned: “(After the attack) there was no negative change (at work). Everyone supported me. Because they knew me very well.” As Participant 2 highlighted, “They know my diagnosis and manage it”; others need to manage this context. Participant 11′s articulation, “Bipolar employees’ views should be recognised; they should not be left out because of their illnesses,” showed the importance of inclusive social settings.(b)Flexible work arrangements: A supportive attitude from supervisors to accommodate the working hours, duration, and space according to the needs of individuals with bipolar disorder was proposed as a critical solution, as the following quotes depict:

“The head of the department handled me in a very fatherly way. He told me not to come to work if I had a problem…. I felt in balance there.” (Participant 15)

“Because my supervisor is very understanding, he knows what my diagnosis is. When I request to go to work late, he gives me an hour to be late. I use my disability rights.” (Participant 7)

“I shared it (bipolar diagnosis) when I started working. I told them that I may need to visit my doctor from time to time. It is not used as a negative thing against me, nor is it used positively.” (Participant 10)

“There are Community Mental Health Centres for patients who need support. Various events are held there. Employees may be allowed to attend the meetings held here. Occupational physicians may write to human resources or the employee’s manager, stating that they must give permission,” (Participant 19)

(c)Allowing for Job Redesign/Replacement: Redesigning or replacing jobs based on individual needs helped maintain employment and prevent unnecessary stressors for employees with BD. Participant 15′s experience set an example for a redesign: “The work-travel was too much. They placed me in a better position afterwards.” Participant 8 described a positive replacement example: “After the attack, my boss supported me. He hired someone while I was gone. After I came back from sick leave, instead of firing him or me, he kept us both.”(d)Allowing for Frequent Breaks: According to participants of the current study, managing stress levels and maintaining mental well-being may easily be promoted by allowing for more breaks. Participant 13 said “When I was working, people were aware (bipolar diagnosis). When I felt the need, I used to step outside the office and be back after a while.”(e)Redesigning Performance Reviews: Redesigning performance reviews involves recognising and valuing the unique contributions of employees with bipolar disorder, emphasising the importance of effective management as Participant 5 portrayed: “When you say to the manager, this employee has a different personality, and we employed them knowingly. How well you manage this person will affect your performance; believe me, that person will stay in business life.”

## 4. Discussion

This study addresses a critical gap in the bullying literature by providing insights into the workplace bullying experiences of individuals with BD and factors that may foster or hinder their participation in the labour force. The study examines the perceived causes and types of negative behaviours, tracking the development of participants’ bullying experiences. Additionally, it evaluates their suggestions for stopping workplace bullying and the accommodations needed to keep them in the workforce for longer periods. All participants expressed a desire to work and maintained a belief in their abilities despite the challenges caused by their diagnosis. The study’s findings indicate that individuals with BD face disadvantages due to their condition, others’ attitudes, and institutional inadequacies in accommodating their needs [42]. This research highlights the need for employers, coworkers, and government institutions to recognise, challenge, and change restrictive behaviours and attitudes towards mental illnesses.

### 4.1. Navigating the Spectrum of Negative Behaviours: Gossip, Ridicule, and Exclusion as Forms of Workplace Bullying towards Individuals with BD

Participants who were exposed to bullying primarily encountered person-related bullying, which included gossip, ridicule, and exclusion, consistent with the existing literature on bullying. Researchers analysing cross-national and cross-cultural similarities and differences in the perception and conceptualisation of workplace bullying among human resource professionals [51] found that in honour-based cultures like Türkiye, where reputation is highly valued, spreading rumours is considered bullying [66]. Early researchers in the field of bullying, who focused on the entire employee base of organisations, defined bullying as a systematic, stigmatising process that violates civil rights [2]. In terms of experiencing exclusion as a form of bullying, individuals with BD are unfortunately at high risk of stigmatisation and exclusion in family, social, workplace, and educational settings. Systematic research on stigma and bipolar disorder indicates that negative behaviour toward individuals with bipolar disorder can be systemic and discriminatory, stemming from beliefs that mental illness signifies personal deficits, weakness, deviance, low intelligence, unreliability, or incompetence, thus hindering social support, functioning, and quality of life [43]. Participants reported experiencing work-related bullying, which included being deprived of work rights, facing unmanageable workloads, and being assigned tasks below their competency level, in line with behaviours mentioned repeatedly by bullying researchers working on a general population of employees [12]. This aligns with previous research on supervisors’ perspective on the challenges involved in job placement for people with mental health problems, indicating that for successful integration of individuals with BD into the workforce, not only must the employee be motivated but the supervisor should also engage in open, continuous communication about the challenges of the diagnosis and work requirements [67]. The absence of such communication can cause individuals with BD to perceive these work conditions as bullying.

### 4.2. Roots of Workplace Bullying: Exploring Organisational Culture, Perpetrator Traits, and Individual Factors in the Onset of Workplace Bullying

First, participants highlighted toxic organisational cultures that allowed for, and did not condemn, bullying behaviours. They mentioned that under the management of absent leaders, bullying flourishes as these leaders are indifferent to employee well-being. Previous reviews on all employee perceptions showed that autocratic and absent leadership styles and the indifference leaders show in resolving work conflicts represent germinating grounds for perpetrators who perceive a lower risk of being caught [68,69]. Participants also pointed out that poor job designs create unmanageable workloads that overwhelm employees, a long-established cause of bullying by researchers working on all employees of organisations, especially when employees have low control over their work [70].

Secondly, most participants identified the negative personal traits of the perpetrators as a major cause of bullying. They characterised these individuals as egoistic, sadistic, opportunistic, mean, unempathetic, and power-hungry. This characterisation aligns with recent empirical studies and systematic reviews, which found that perpetrators often exhibit high levels of sadism, Machiavellianism, narcissism, and psychopathy, traits that lead them to be perceived as manipulative, dishonest, insincere, and lacking in empathy [9,71].

Participants predominantly experienced bullying from their supervisors and peers. This is consistent with findings from literature reviews [49], especially in high-power distance countries like Türkiye [50,51].

Participants noted that their behaviour may be one of the causes of being bullied. Research indicates that pre-existing physical or mental illness may raise an individual’s risk of being bullied [13]. Participants mentioned that they have distinct behaviours when they are depressed or euphoric. When in depressed episodes, individuals with BD may attract perpetrators, as they may appear depressed, scared, sad, anxious, and unable to retaliate [69]. On the other hand, when euphoric, employees diagnosed with BD may have higher energy levels than usual and disturb their colleagues because they are more self-confident [72].

Despite work-life benefits, there are also downside risks to being in work-life for individuals with mental illnesses. Although they may have unique skills and qualities that enable them to succeed in their jobs, they also tend to overwork [73], intensely feel the pressures of responsibility, and may have angry outbursts [74]. Resultantly, they may also have elevated stress levels and reduced abilities to cope with it [75], attracting perpetrators.

### 4.3. The Devastating Impact of Workplace Bullying on Employees with BD: Ignored Complaints and Health Consequences

Bullying creates stress, incapacitating the individual, exacerbating the diagnosis, creating a downward spiral for employees with BD [20,72], and leading to depressive and manic episodes. A previous systematic review of bullying outcomes over all employees of organisations showed that exposure to bullying is linked to eroding mental and physical health, post-traumatic stress symptoms, increased leave intentions, and decreased job satisfaction [76]. The present study adds to bullying literature that bullying triggers bipolar episodes. While experiencing bullying, participants indicated that they mostly felt sad and ashamed.

Most participants did not file a complaint while experiencing bullying as targets in high-power distance cultures like Türkiye may tolerate negative behaviours [50,77]. However, bullying did not stop for years, even for those who have filed complaints. In line with the results of a previous study, when experiencing bullying, participants did not mention getting help from their healthcare team, including occupational health physicians or occupational health and safety committees at work [78].

The risk of experiencing bullying is higher for individuals with BD than employees with an anxiety disorder or depression, according to research conducted involving employees with mental illnesses [79]. Research on the entire workforce showed that targets might be caught up in the vicious cycle of bullying due to worsening mental health [15,76]. Most participants were passively coping and felt powerless against bullying, which may have encouraged their victimisation. Negative feelings accumulated, and interpersonal conflicts remained unsolved as these employees were left alone in their toxic environments, as established by previous researchers working with the entire workforce of organisations [5].

Eventually, most participants experiencing bullying resigned, as it is the most common consequence of bullying for targets and victims, irrespective of mental health status [64]. Other times, participants continued to endure bullying to keep their jobs. Some employees tolerated continued negative behaviours for fear of being unable to find a job again, further jeopardising their health. Diagnosis-related dismissal from their jobs was “the most inhumane act” of all, as mentioned by Participant 1. None of the participants initiated legal action against their employers due to fear of receiving negative references for future job searches and potential material gains (e.g., a four-month salary) being too low to compensate for the stress it would create for the individual.

Participants mentioned that HRPs actively fired, demoted, or even insulted them because of their diagnosis, sided with management, or failed to stand up for the rights of this group. A previous study from the viewpoint of HRPs in Türkiye showed that behaviours such as teasing someone and assigning excessive workloads are considered acceptable in modern business life. Furthermore, if the perpetrator significantly contributes to the organisation’s job performance, these negative behaviours may be disregarded, indicating a potential bias in addressing workplace bullying based on individual performance contributions [77]. Participants seemed to consider HRPs as mainly ignorant of their diagnosis, unskilled in disclosure procedures, and unwilling to provide accommodations at work.

The results aligned with the findings from cross-cultural comparisons concerning workplace bullying. Researchers [50] compared Australian, Indian, and Turkish individuals’ target experiences. In Turkish culture, which emphasises collectivism, respect for hierarchy, and harmony, individuals reported a higher tolerance for negative behaviours, often attributing blame to themselves. They tended to experience downward bullying and to avoid reporting incidents or seeking assistance from unions. Despite these cultural nuances, the fundamental experiences of workplace bullying showed striking similarities worldwide, reflecting common elements in human life and nature, as well as the impact of globalisation. In a different cultural comparison study [80], Türkiye was categorised among survival cultures that prioritise physical and economic security. This focus leads to an obsession with controlling tasks and obligations and a general mistrust of others. Such cultural characteristics are associated with a higher prevalence of employee harassment.

In conclusion, individuals with BD who needed jobs to preserve their well-being were bullied and suffered even worse health conditions. Top management, who were legally responsible for their well-being, ignored and dismissed their complaints.

### 4.4. Towards a Bully-Free Workplace: Empowering Strategies for Individuals and Organisational Change

Experiencing bullying is devastating for all employees’ psychological and physical health, and employees with BD are even more brutally impacted. Participants urged organisations to employ leaders who care about the well-being of employees, to establish and announce policies and procedures to stop bullying, and to train employees on how to approach employees with mental illnesses. They also urged individuals with BD to be honest about their diagnosis, to educate coworkers, and to employ various coping mechanisms against bullies.

These suggestions are aligned with those from a systematic review of research conducted with employees, irrespective of their mental health status. This review recommended organising training on conflict management and anti-bullying; allowing for job crafting; being attentive to conflicts; discouraging conflict-escalating behaviours; setting clear accountability; and introducing employee health and safety rules [71].

Participants also urged governments to arrange public scanning and training on mental illnesses, to audit organisational conduct against employees with mental illnesses, and to revise the law on bullying to protect the victims. Some participants suggested becoming self-employed or working for public organisations to lower the risk of discrimination and bullying.

### 4.5. Building a Supportive Ecosystem: Accommodations for Prolonged Employment and Psychological Well-Being

For a long time, the general prejudice against psychiatric illnesses, including BD, prevented this group from being in the workplace, and the literature focused on the cost aspect of mental illnesses for organisations [81,82]. However, a recent systematic review on work accommodations for this population revealed that accommodations improved the length of job tenure, reduced the severity of certain mental disorders, and the costs associated with these accommodations were found to be minimal, having positive economic benefits for employers [83].

All participants in this study mentioned that they are productive, hardworking individuals with creative intelligence and a variety of skills. They emphasised that they are often interested in life-long learning and continued their educational pursuits despite the disruption caused by their illness. Participants who accepted the diagnosis and received uninterrupted treatment were able to continue their education, extending their working life and replicating the results of a previous study on employees with bipolar disorder [32]. In terms of organisational factors, in line with previous research, the most crucial factor for their longer-term employment was having a supportive organisational culture, wherein management and colleagues create a safe environment for them to share their diagnosis, and in which they feel accepted and valued [84]. Several participants reported that the stigma associated with BD has led them to be fired from jobs, passed over for promotions, or demoted right after disclosure, as previously reported by other researchers working with employees diagnosed with bipolar disorder [31,78]. HRPs seemed to lack knowledge of BD, disclosure processes, and accommodations. Some managers did not trust HRPs to handle these procedures in favour of the employee and therefore kept the diagnosis confidential from HRPs.

Participants mentioned that prescribed medications cause insomnia, excessive sleep problems, attention deficits, memory problems, absenteeism, presenteeism, and slow work pace. Therefore, taking frequent breaks and leave of absence if they feel unwell, having job security with flexible work schedules [85], and being able to shift to less stressful positions [78] were important factors in maintaining their mental health in balance.

Other necessary accommodations, not mentioned by the study participants but established by previous studies on employees diagnosed with bipolar disorder, included assistance from mental health professionals [86], occupational safety and health physicians for objective assessment of mood changes [87], and job coaches for help in cognitive hardships and interpersonal skills [88,89] and job placements [67].

### 4.6. Transformative Steps: Addressing Stigma, Enhancing Awareness, and Fostering Inclusive Work Environments for Individuals with BD

This study shows that individuals with BD are stigmatised, discriminated against, and bullied at work, not only by their colleagues but also by company owners and HRPs. They do not have the entire support systems present in Western cultures. Our results have three main practical implications.

First, workplace bullying can trigger BD episodes, potentially shortening employment duration and exacerbating well-being issues for those with BD. Given the associated risks of suicidal ideation and behaviour [90] and higher suicide risk in people with BD [38], it is crucial for employers to protect these employees. Enhancing awareness through occupational health professionals and labour unions is essential.

HR professionals often overlook social exclusion, which is a significant form of discrimination against individuals with BD. Research shows that in high in-group collectivist cultures like Türkiye, social isolation is not commonly recognised as bullying [51], yet subtle forms of discrimination are as harmful as overt types [91]. This underscores the need for diversity, equity, and inclusion training in organisations, as encouraged by the European Commission [92].

To address workplace bullying, organisations must first tackle the stigma associated with mental illnesses. Diversity training, particularly in collectivist cultures [51], should focus on the negative impacts of stigma and the importance of respect for mental health issues [93]. Further steps include team-building exercises to foster group cohesion [94,95], establishing a vision that welcomes employees with mental illnesses and promoting an inclusive work culture [96,97,98]. Anti-discrimination and bullying policies should be implemented, with management actively supporting these initiatives [51]. It is also crucial to facilitate the disclosure process for those with mental illnesses, considering their anticipated discrimination [98].

As a broader scale intervention, policymakers, social services, universities, and non-profit organisations could hold conferences and training sessions for the management and HRPs of organisations to educate them on mental illnesses, how to handle disclosure processes, and how to set accommodations for the long-term employment of individuals with BD. These events may increase awareness and trigger a more inclusive climate in organisations.

### 4.7. Limitations

Due to the limited sample size, the results should be interpreted cautiously as they do not represent all individuals with BD. One may argue that our sample size of 19 is relatively small compared to other qualitative studies conducted with individuals with BD (*n* = 35, [99]; *n* = 22, [40]; *n* = 35, [78]), but we have a very diversified sample in terms of age, gender, occupation, and education level. In qualitative surveys, it is not population representativeness that is sought but rather a diversity of situations [100]. As the sample was recruited from the Bipolar Individuals group page on Facebook, possibly high-functioning, relatively high- to middle-status employees were in the group, who may have reported low occurrence of, and impact of, negative effects of bullying. Lower-status occupations may have more problematic working conditions affecting this population, such as physical demands, long or shift work hours, and heavy supervision [74]. The negative behaviours mentioned in this study were not measured by administering standard scales, but the scale items were matched with their descriptions. In addition, these statements were not cross-examined with witnesses, perpetrators, managers, and HRP for validation. The study’s reliability might have been improved through participant validation. Recall bias was also a limitation as subjects were asked to recall bullying situations, sometimes going back ten years. Finally, this study did not specifically investigate and rule out other comorbidities associated with bipolar disorder. Participants were not explicitly asked about other mental health conditions they might have, such as post-traumatic stress disorder or borderline personality disorder, which significantly influence emotional sensitivity and cognitive perception in people with bipolar disorder. However, a few participants did disclose their condition of anxiety disorder.

### 4.8. Future Research

While medical sciences have been studying individuals with BD, research in the social sciences on the work conditions of individuals with BD has, globally, lagged behind considerably. Future research could be longitudinal or involve diary studies examining daily or weekly work conditions, their impact on behaviour, and the need for accommodations to identify problematic work conditions versus those helpful in managing symptoms of this illness. Another future direction would be to focus on specific job categories, such as blue- or white-collared employees or company sizes, to refine our understanding of the implications of the bullying experienced and the working accommodations specific to people with BD. The research on individuals with BD could also be expanded to cover all employees with different mental illnesses (e.g., psychosis, schizophrenia) and neurodivergent employees. Future research could also include the viewpoints of policymakers, unions, top management, occupational physicians, occupational health and safety committees, and HRPs on the successful inclusion of accommodations for employees with mental illnesses.

## 5. Conclusions

Despite their constant battle with BD, this population wants to work, accomplish goals, and feel they belong. Stigma and bullying of individuals with BD are significant impediments to the well-being of this group. Workplace bullying, at its core, shares numerous identical aspects across the employee base but shows its detrimental side with this population by triggering manic-depressive episodes. Supervisors are mentioned as the most active perpetrators, while HRPs are falling far short in protecting the well-being of these individuals. The causes of bullying were cited as perpetrators’ undesirable personalities, participants’ mood states, poor leadership, and organisational structures where negative behaviours are ignored, allowed, and not punished. Public education and awareness campaigns on mental illnesses and workplace bullying, mirrored in organisations, could be the first step in developing efficient interventions to tackle the bullying and discrimination of individuals with mental health illnesses. Businesses could promote a healthy work environment and support wellness among all employees by lowering unnecessary workplace stress. They could consider implementing anti-stigma initiatives that promote hiring people with mental disorders into the workforce. Keeping individuals with BD in the workforce is a challenging task. Policymakers, top management of organisations, and HRPs could prevent punitive conduct and build supportive organisational cultures for their inclusion and well-being. All these steps will enable organisations to provide a meaningful work life free from bullying for the individuals in question, one that aligns with their educational accomplishments, expectations, skills, and goals, enhancing their well-being and positively impacting society.

## Figures and Tables

**Figure 1 ijerph-21-00137-f001:**
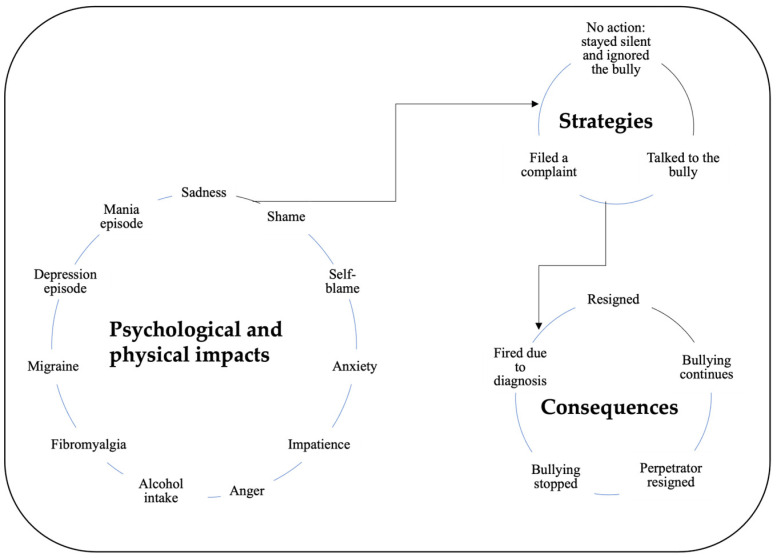
Concept map of bullying’s impact, coping strategies, and consequences.

**Table 1 ijerph-21-00137-t001:** Participants’ sociodemographic profiles.

#	Gender	Age	D. Age	Exp. (Years)	# of Comp.	Education	Last Sector Worked	Job Title	Currently Working	Exposed to Bullying	Diagnosis Disclosed
1	F	39	32	17	4	GD-cont.	Manufacturing	Strategic planning manager	Yes	Yes	Yes
2 *	M	30	16	13	1	Pre-UD	State org.	Clergyman	Yes	Yes	Yes
3 *	M	33	28	17	4	UD	Retail	Department manager	Yes	No	Yes
4	M	52	45	26	4	UD	Manufacturing	Sales, marketing, and administration	No-seeking	Yes	No
5	F	55	55	26	4	UD	Manufacturing	Training manager	No-retired	Yes	No
6	M	26	16	1	1	UD	State org.	Data preparation and control	Yes	Yes	Yes
7	F	32	14	2	2	UD	State org.	Social worker	Yes	No	Yes
8	M	35	29	11	3	GD	Retail	Dealer manager	No-seeking	Yes	Yes
9	M	23	19	4	1	UD	NGO (Student)	Head of Council of Disabled Students	No-seeking	Yes	Yes
10 *	M	30	17	5	3	UD-cont.	Retail	Office Manager	Yes	Yes	Yes
11	M	34	34	8	1	GD	State org.	Project manager	Yes	Yes	Yes
12	M	41	21	21	NA	JH	Manufacturing	Construction worker	Yes	No	Yes
13	F	50	40	20	NA	JH	Manufacturing	Sewing machine operator	No-retired	No	No
14	F	25	23	6	3	UD	Media	Designer	Yes	Yes	No
15	F	37	19	16	3	GD	State org.	Coordinator	Yes	Yes	Yes
16	F	39	29	17	2	GD	State org.	Food engineer	Yes	No	No
17	M	35	28	10	6	Pre-UD	Exports and Imports	Business owner	No	No	No
18	F	27	24	1	1	Pre-UD	Media	Business admin student	No	Yes	No
19	M	34	32	10	3	UD	State org.	Market research specialist	Yes	Yes	Yes

Notes: * Face-to-face interview participants. Exper. (years): Lifetime work experience; D. Age: Diagnosis age; # of comp.: Number of companies worked; GD: Graduate degree; UD: Undergraduate degree; cont: education continued; JH: Junior High; NA: Missing data; NGO: Non-governmental organisation; State org.: State organisation.

**Table 2 ijerph-21-00137-t002:** Bullying experiences of individuals with BD.

Types	Negative Acts *	Sample Quotes
Person-related bullying		
Being insulted		Behind my back, they (colleagues) said, “is she retarded or crazy.” (Participant 18)
Being ignored or excluded		They excluded me because of my depression. (Participant 5)
Persistently criticised		People constantly criticised my personality, not how I did my job. (Participant 5)
Excessively teased with sarcasm		It (diagnosis) was a matter of constant teasing behind my back. (Participant 18)
Work-related bullying		
Unmanageable workload		Unfortunately, in Turkey, the lack of clear working hours is very exhausting. Working under non-ideal conditions, with very long working hours and standing for extended periods, was very difficult, especially for someone like me. Did you also work on weekends? Yes, especially in real estate, we used to work on weekends and take one day off during the week. Potential buyers would visit on weekends. When you work for 10–11 h a day, trying to complete tasks that you couldn’t finish during the week, life becomes unbearable. This is very difficult even for healthy individuals, and for a bipolar person, it is much, much more challenging. Because I need to be able to clear the energy I put into my own situation. When I can’t do that, it becomes impossible to move. Participating in life becomes difficult. (Participant 10).
Deprived of work rights		I was having trouble getting permission for annual leave. We could not use it to the end when we were given permission. (Participant 19)
Working below level of competence		(When the CEO and HR learned about my diagnosis) They said, ‘You are an unstable person. You cannot manage a team. “You are not a manager anymore”, and they downgraded me. (Participant 1)
Physically intimidating bullying		
Being shouted at		When he suddenly enters the room shouting (supervisor), it feels like our hearts will burst (Participant 11)

*** Coding themes based on NAQ-R [60].

**Table 3 ijerph-21-00137-t003:** Causes for workplace bullying.

Themes	Example Quotes
Work environment	
Toxic environment	“I was excluded from others in my first job by my peers since I was a relative of the owner of the company.” (Participant 8)
	“Bullying others is a learned behaviour. This is how they learn from others. They continue to practice it unconsciously or knowingly, like a tradition.”(Participant 11)
	“I was trying to get a disability report. Later, I found out that this was the subject of teasing in the company.” (Participant 18)
Leadership	“If leadership is inadequate, anything could happen.” (Participant 4)
	“I had conflicts with my manager on business issues. His comments were soul-crushing.” (Participant 19)
Workload	“There was trouble on task distribution since the workload was too much.” (Participant 6)
	“He was constantly putting pressure on us to finish the task at hand. We were constantly stressed. He was verbally abusive.” (Participant 11)
Individual Characteristics	
	“Ego problems” (Participant 6)
Related to the bully	“People are ruthless and mean” (Participant 18)
	“It is a power game” (Participant 2)
	“If the managers are not very confident, feel someone as a threat, know that an employee under them is actually better than them, do not want that person to come to the forefront, then they are targeted.” (Participant 15)
	“Managers have encountered such an attitude previously. So it is a chain thing. It is a learned behaviour. That is how they conduct their relationship with others. They continue with negative behaviours unconsciously or knowingly, like a tradition in the organisation. They probably also bully their spouses.” (Participant 11)
Related to the victim	“If I am unhappy, I am unhappy. I cannot fake it, but everybody else is fake. We are different from them.” (Participant 9)
	“You are already aware that you are different. Some people just cannot handle that difference. They are intimidated. When they meet a colourful personality, they see you as a threat in some way.” (Participant 10)

## Data Availability

We are not allowed to share data, due to consent statements.

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
