# Peer review of "Is There Somebody Looking out for Me? A Qualitative Analysis of Bullying Experiences of Individuals Diagnosed with Bipolar Disorder"

_ijerph, 2024, doi:10.3390/ijerph21020137_

Round 1

Reviewer 1 Report

Comments and Suggestions for Authors

This research article explored the experiences of employees with bipolar disorder (BD) in Türkiye, focusing on workplace bullying (WB) and its impact on their well-being and work performance. The study utilized in-depth interviews with 19 individuals diagnosed with BD.

The research methodology, including the number of participants, data collection tools, and analysis methods, are clearly outlined. The use of in-depth interviews and content analysis is appropriate for exploring the personal experiences of participants

However, it would be beneficial to include a discussion of workplace bullying experiences among individuals who do not have mental health disorders, as per existing literature. This addition would help address the limitation related to the lack of a control group, offering a valuable point of comparison and augmenting the study's comprehensiveness.

In conclusion, this article provides valuable insights into the challenges faced by individuals with bipolar disorder in the workplace.

Author Response

Please find attached our responses to Reviewer 1. 

Thank you 

Best regards

Reviewer 2 Report

Comments and Suggestions for Authors

The topic very interesting and unique, and I agree that it is a good piece of work. However, I also acknowledge that there are a few areas that need editing. 

Here are some of the suggested edits: - The introduction regarding bipolar disorder could be clearer and more informative. It would be helpful to explain the criteria using either DSM 5 or ICD-11. - The study did not mention ruling out PTSD or borderline personality disorder, which can be important factors in emotional sensitivity and cognitive perception in people with bipolar disorder. - Providing more insight into the culture and dynamics in the study would be beneficial, since the study was conducted in Turkiye, which has a different culture and attitude in the workplace. - Including a controlled group with normal participants, as well as considering confounding variables such as workload, personality issues, and peer/supervisor dynamics, would enhance the study's validity. - Sections 338-345, 355-360, and 362-365 need improvement, as there is too much repetition and the statements are not very convincing. - The section on practical implications is lengthy and could be shortened. - The limitations of the study include the lack of a control group, a small sample size, and the failure to rule out PTSD and borderline personality disorder in participants. - It is worth noting that some of the examples and quotes provided in the study suggest that the patient is suffering from borderline personality disorder. For instance, example quote 3 by participant 15 in table 4 indicates emotional sensitivity, which is a common trait in people with borderline personality disorder. Therefore, it would have been helpful if the study had ruled out borderline personality disorder in participants. I hope this revised version meets your expectations. If you have any further suggestions, please do not hesitate to let me know.

Comments on the Quality of English Language

Good. Minor revisions needed 

Author Response

Please find attached our responses to Reviewer  2. 

Thank you 

Best regards

Reviewer 3 Report

Comments and Suggestions for Authors

Thank you to have the opportunity to revise this paper including a qualitative analysis of bullying experiences of individuals diagnosed with bipolar disorder. Although the argument could be of interest in patients with bipolar disorder, the manuscript have strong limitations:

1. the design: 

2. the sample consists of only 19 patients. Usually the small sample, tipically of case series, are used for something of very innovative or biologically interesting, or investigating neuroimaging. Furthermore, they are used for very very omogeneous patients, to characterize phenotypically. 

3. the interview: on 19 patients, 16 patients were interviewed by telephone.

For these reasons, i suggest to not proceed further to the revision of manuscript.   

Author Response

Please find attached our responses to Reviewer 3. 

Thank you 

Best regards

Round 2

Reviewer 3 Report

Comments and Suggestions for Authors

Although the argument could be of interest in patients with bipolar disorder, the manuscript have strong limitations as described in a previous revision process. Therefore, i suggest to not proceed with the publication. 

Author Response

Dear Reviewer

Thank you for your comments. We have strived to address all your concerns and have added explanations to the manuscript. We hope it will meet your acceptance this time.
